# Tree height strongly affects estimates of water-use efficiency responses to climate and $CO_2$ using isotopes

R.J.W. Brienen [1], E. Gloor[1], S. Clerici[1], R. Newton[2], L. Arppe[3], A. Boom[4], S. Bottrell[2], M. Callaghan[1], T. Heaton[5], S. Helama[6], G. Helle [7], M.J. Leng[5,8], K. Mielikäinen[9], M. Oinonen [3] & M. Timonen[6]

Various studies report substantial increases in intrinsic water-use efficiency ($W_i$), estimated using carbon isotopes in tree rings, suggesting trees are gaining increasingly more carbon per unit water lost due to increases in atmospheric $CO_2$. Usually, reconstructions do not, however, correct for the effect of intrinsic developmental changes in $W_i$ as trees grow larger. Here we show, by comparing $W_i$ across varying tree sizes at one $CO_2$ level, that ignoring such developmental effects can severely affect inferences of trees' $W_i$. $W_i$ doubled or even tripled over a trees' lifespan in three broadleaf species due to changes in tree height and light availability alone, and there are also weak trends for Pine trees. Developmental trends in broadleaf species are as large as the trends previously assigned to $CO_2$ and climate. Credible future tree ring isotope studies require explicit accounting for species-specific developmental effects before $CO_2$ and climate effects are inferred.

---

[1] School of Geography, University of Leeds, Leeds LS6 9JT, UK. [2] School of Earth and Environment, University of Leeds, Leeds LS6 9JT, UK. [3] Laboratory of Chronology, Finnish Museum of Natural History-Luomus, University of Helsinki, PO Box 64, 00014 Helsinki, Finland. [4] School of Geography, University of Leicester, Leicester LE1 7RH, UK. [5] NERC Isotope Geosciences Facilities, British Geological Survey, Nottingham NG12 5GG, UK. [6] Natural Resources Institute Finland, Eteläranta 55, PO Box 16, 96301 Rovaniemi, Finland. [7] GFZ — German Research Centre for Geosciences, Section 5.2 Climate Dynamics and Landscape Evolution, Telegrafenberg, 14473 Potsdam, Germany. [8] Centre for Environmental Geochemistry, University of Nottingham, Nottingham NG7 2RD, UK. [9] Natural Resources Institute Finland, Jokiniemenkuja 1, PO Box 18, Vantaa 01301, Finland. Correspondence and requests for materials should be addressed to R.J.W.B. (email: r.brienen@leeds.ac.uk)

The Earth's vegetation is an integral part of both the global hydrological cycle and the global carbon cycle, annually transpiring more than twice the amount of water vapour in the atmosphere[1], and processing ca. 120 Gt carbon through photosynthesis[2, 3]. Changes in the functioning of Earth's vegetation therefore affect these cycles and future climate change[4], and are also a concern on their own. There is thus great interest in understanding plant responses to recent human-induced changes in physical conditions at the Earth's surface. Recent increases in atmospheric $CO_2$ concentrations, nitrogen deposition and rising temperatures all potentially affect plant growth. The increase of atmospheric $CO_2$ in particular is expected to benefit growth, as it facilitates the uptake of carbon, and potentially leads to an increase in plant water use efficiency. Water use efficiency is defined as the amount of carbon gained by plants per unit water lost[5]. Such responses have indeed been observed in $CO_2$ enrichment experiments in greenhouses and under natural conditions[6], in ecosystem flux studies[7] and in tree ring and foliar carbon isotopes studies[8–12]. However, the magnitude of measured responses varies between methods, and thus large uncertainty remains with regard to the magnitude of water use efficiency changes over time.

A popular method of studying plant water use efficiency is to use the carbon isotope composition of tree rings ($\delta^{13}C_{plant}$, see Methods section). It is attractive because it provides long-term annual records. From tree ring $\delta^{13}C$ and historical records of atmospheric $\delta^{13}C$, plant isotope discrimination ($\Delta^{13}C_{plant}$) can be calculated (see Methods section), which provides an estimate of changes in the ratio between assimilation and stomatal conductance for water vapour ($A/g_w$). This ratio is called intrinsic plant water use efficiency ($W_i$). Most studies which perform trend analysis of past change in $W_i$ use tree cores from large trees, which inherently include all developmental effects within individual trees. Typically, these studies find increases in trees' intrinsic water-use efficiency in all biomes in the order of 10–30% over the past 150 years[5, 8, 9, 13–22].

As already mentioned, common to most of these studies is the implicit assumption that developmental and stand level environmental effects on $W_i$ over a trees' lifespan are negligible. However, various existing studies cast doubt on this assumption.

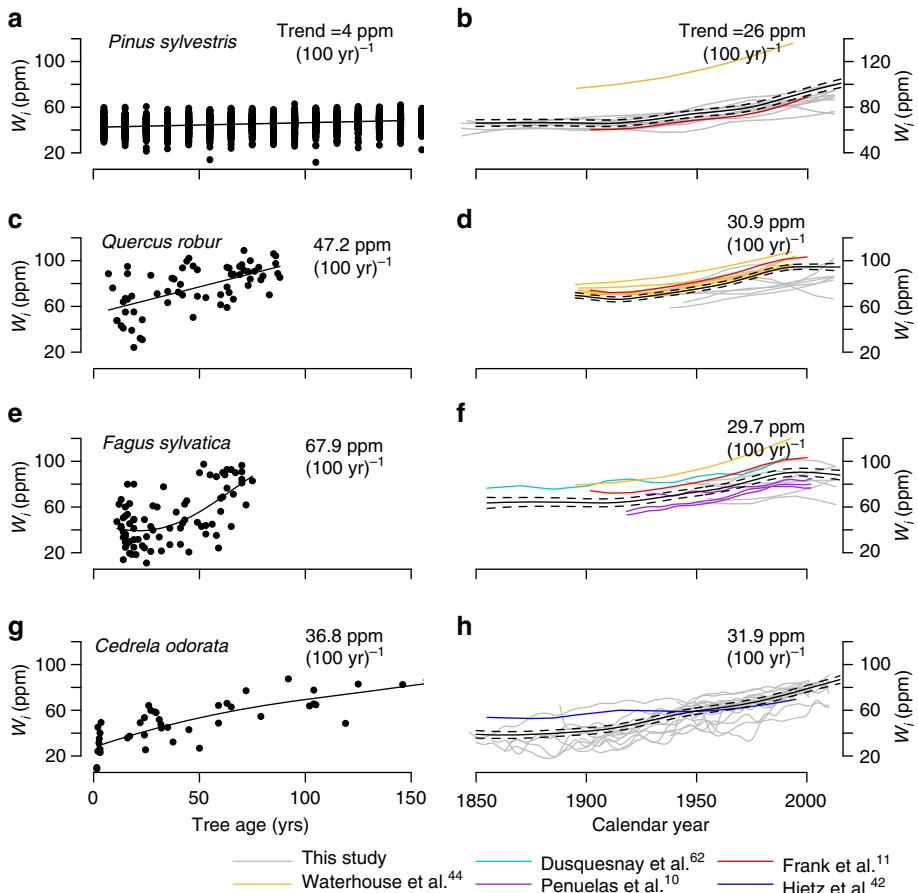

Fig. 1 Developmental trends and time trends of intrinsic water use efficiency for four species derived from tree ring carbon isotopes. Developmental trends (a, c, e and g) are controlled for variation in $CO_2$ by using sub-fossil trees growing under relatively constant pre-industrial $CO_2$ (for Pinus), or by plotting $W_i$ derived from the five outermost rings across a range of trees differing in tree age (for Quercus, Fagus and Cedrela). Time trends (b, d, f and h) are reconstructed based on classic tree ring approaches, which infer past change in $W_i$ using pooled or individual tree ring $\delta^{13}C$ series derived from big trees when these were younger and smaller, thus assuming negligible developmental effects. The data for Pinus consist of 10-year time series from 182 sub-fossil trees (5630 BP–AD 1930) from northern Fennoscandia (Helama et al.[39]). Time trends in $W_i$ are estimated using published trends (coloured lines), complemented with new data collected from dominant trees in this study (grey lines, see Methods section). Black lines show mean trend estimates using general additive mixed models (GAMM) with standard errors (broken lines). Long-term linear $W_i$ trends were estimated using linear mixed-effects models with weighting for sampling intensity for each study (see Methods section). Note that we excluded trees with size lower than 1 m from the age trend analysis here (a, c, e and g) to avoid influence of soil-respired carbon. Included literature data are from Frank et al.[11], red, Waterhouse et al.[46], yellow, Duquesnay et al.[62], cyan, Penuelas et al.[10], purple and Hietz et al.[45], blue (for details see Supplementary Table 1)

**Table 1 Analysis of age- and height-related changes in $W_i$ for different life stages**

| Species | $W_i$ vs. age trends ppm $(100\ yr^{-1})$ | | | | | |
|---|---|---|---|---|---|---|
| | Full age range | Age > 25 yrs | >50 yrs | >75 yrs | >100 yrs | >200 yrs |
| *Pinus sylvestris*[a] | 3*** | 2.9*** | 2.4*** | 2.2*** | 2.0*** | 2.6** |
| *Quercus robur* | 43.1*** | 27.7* | 65.3* | | | |
| *Fagus sylvatica* | 69.5*** | 99.4*** | 185* | | | |
| *Cedrela odorata* | 39.1*** | 26.1*** | 14.3* | 15.2$^{NS}$ | 12.9$^{NS}$ | |
| | $W_i$ vs. height trends, ppm $(10\ m)^{-1}$ | | | | | |
| | Full height range | Height > 1 m | >2.5 m | >5 m | >10 m | >20 m |
| *Pinus sylvestris*[b] | −0.41$^{NS}$ | 1.74 | 8.42* | 9.94* | 14.7 | |
| *Quercus robur* | 17.43*** | 20.22*** | 20.84*** | 14.94*** | 13.95** | |
| *Fagus sylvatica* | 18.41*** | 18.08*** | 18.16*** | 23.04*** | 26.10*** | 52.28** |
| *Cedrela odorata* | 19.37*** | 17.28*** | 15.81*** | 16.27*** | 15.72*** | 16.10** |

Linear trends in $W_i$ vs. age or height for full range of data, and excluding earlier life stages (e.g., first 25 years, 50 years, etc, or first metre, first 2.5 m, etc). Results show that age trends in *Quercus* and *Fagus* remain strong, while *Cedrela* and *Pinus* show slight decreases in the slope of the trends with age. Trends with tree height remain strong throughout the full height range Significance levels are indicated as follows: NS not significant; $*p < 0.05$; $**p < 0.01$; $***p < 0.001$
[a]Using the data set of sub-fossil pine trees from Helama et al.[39]
[b]Using the data set of *Pinus* trees with height measurements from the UK (see Methods section)

These studies show that plant isotope discrimination (and thus $W_i$) changes strongly as trees grow in height (McDowell et al.[23] and refs therein), giving rise to an age effect[24] that may confound the interpretation of $W_i$ in terms of climate or $CO_2$. This phenomenon is also called the 'juvenile effect'[25] or 'canopy effect'[26], and will be further referred to as 'developmental effects'.

That there are developmental effects is not very surprising. First, as trees grow older they increase in height, which imposes gravitational constraints on water transport to leaves in the upper canopy, affecting potentially stomatal conductance[27–29]. In addition, trees in closed-canopy forests experience strong increases in irradiance from the understory to the canopy, which affect rates of photosynthesis[30]. These changes in stomatal conductance and photosynthesis as trees grow both affect plant isotope discrimination[23, 31]. Associated developmental changes in tree morphology and physiology may further affect isotope discrimination[28, 32–35]. Yet another factor that influences tree $\delta^{13}C$, and thus estimation of $W_i$ based on isotopes, is the uptake of soil-respired $CO_2$ by trees growing close to the forest floor[26, 36]. As respired soil carbon is dead organic plant material, it increases the $CO_2$ concentration in air above the forest floor and lowers $\delta^{13}C_{air}$ of $CO_2$ in the air[37], potentially affecting the isotope signal for small trees.

Despite the results of these studies, with obvious implications for the interpretation of tree ring-derived time trends in $W_i$, not many studies have explicitly assessed the magnitude of these effects (but see Marshall and Monserud[38]). A possible method to study developmental effects is the use of old fossil trees that grew their entire life under more or less constant $CO_2$ and climate[39]. Unfortunately such ancient trees are rare. An alternative method is to sample trees across all size classes at the same $CO_2$ level at a single site (i.e., size-stratified sampling), which in addition allows to disentangle the specific causes behind developmental changes in $W_i$. The few studies that followed this approach show that tree ring-derived $W_i$ increases may be strongly overestimated when neglecting age effects[38, 40]. However, only a few single species have been studied, and it is still quite unclear to what degree developmental trends affect $W_i$ in the majority of the species used in isotope dendrochronology studies. It also remains unclear to what extent these developmental effects are indeed just simply related to tree age, or rather caused by a change in tree height, a changing light environment of trees, or uptake of soil-respired $CO_2$ when growing under the canopy. We also do not know

whether developmental effects halt at a mature tree stage, an assumption made by several studies attempting to reconstruct historical water use efficiency[41–43]. Therefore, here we collect new data using a systematic size-stratified approach that controls for $CO_2$ to quantify, and unravel developmental effects on $W_i$ in three species commonly used in temperate zone isotope studies (*Pinus sylvestris*, *Quercus robur* and *Fagus sylvatica*) and one representative of tropical species (*Cedrela odorata*). For *Pinus*, we furthermore study the change of $W_i$ over the lifetime of individual trees from sub-fossil trunks from Finland. These have been recovered from lakes and grew their entire life under more or less constant pre-industrial $CO_2$ levels[39]. Altogether, our data set encompasses a variety of ecosystems (arctic, temperate and tropical forests), time-scales (recent to sub-fossil, Holocene material) and species.

The specific aims of this paper are to assess the magnitude of changes in $W_i$ across different life stages of trees, to identify the drivers behind changes in $W_i$ over a trees' life cycle, including tree height, light, age and soil respiration effects, and to discuss the implications of these developmental changes in $W_i$ on estimates of responses to atmospheric $CO_2$. The basic method we use to evaluate the effect of tree developmental changes on $W_i$ is to compare $W_i$ of the last five rings formed under nearly the same atmospheric $CO_2$ conditions from a range of trees varying in age, size and light availability. While this approach does not fully control for each of these co-varying factors, the relatively large sample size does allow for separation of the various effects on $W_i$. To assess to what degree observed trends in $W_i$ may be affected by trees' ontogeny, we juxtapose the developmental changes in $W_i$ with observed time trends. We further use the isotope data to discuss lifetime changes in the context of proposed strategies for gas-exchange regulation[11, 12, 17].

We find that developmental trends in $W_i$ are very strong in the three broadleaf species, doubling or tripling over a trees' lifetime. These trends are primarily caused by increases in tree height and changes in light environment. Trends are of comparable magnitude to observed time trends in literature questioning the interpretation of these records as $W_i$ responses to climate and $CO_2$.

## Results

**Age and time trends in $W_i$.** The three broadleaf species, *Quercus*, *Fagus* and *Cedrela*, exhibit strong increases in $W_i$, doubling

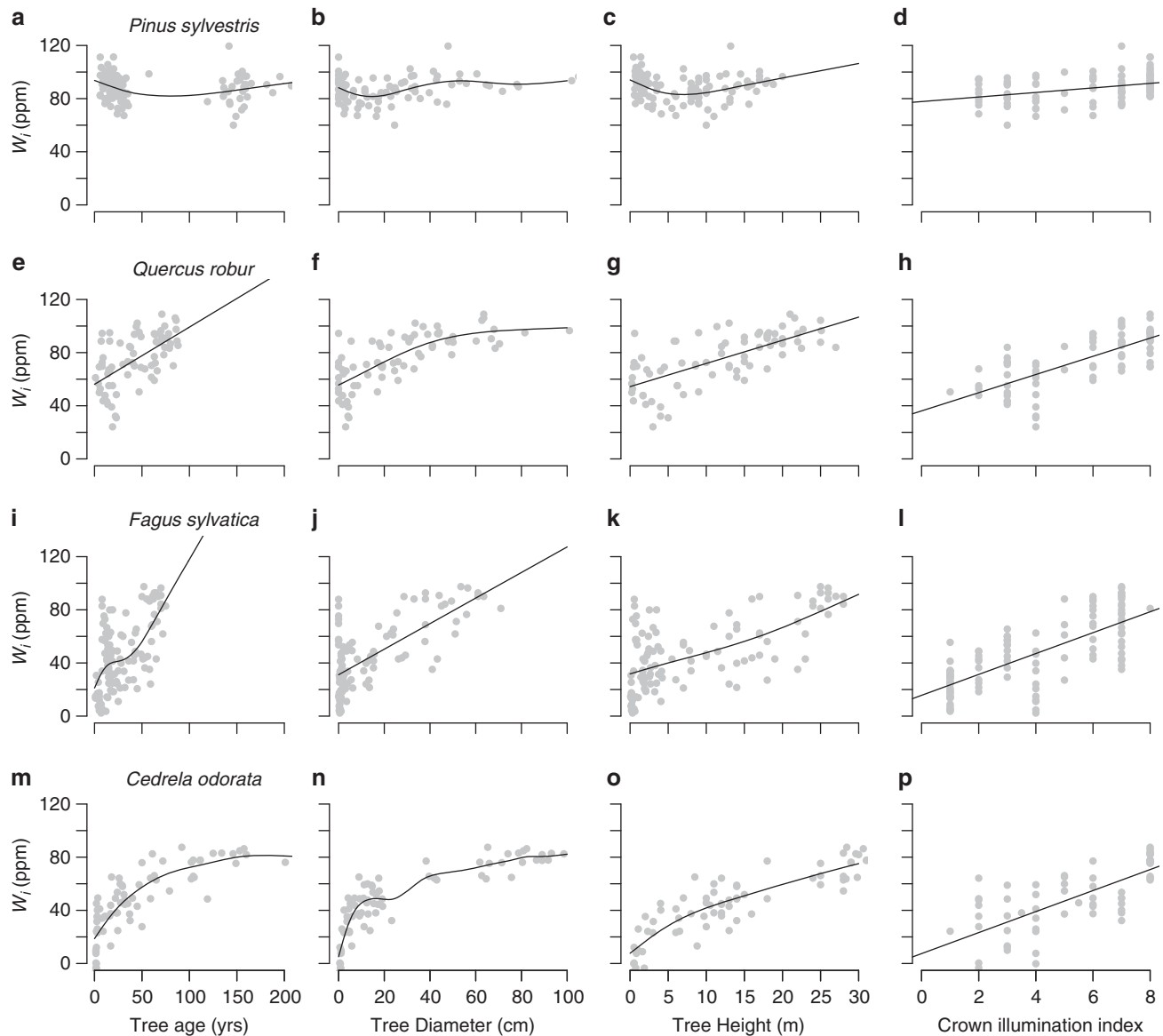

**Fig. 2** Relationship between intrinsic water use efficiency and tree age, diameter, height and crown illumination for the four species. **a–d** shows *Pinus sylvestris*, **e–h** shows *Quercus robur*, **i–l** shows *Fagus sylvatica* and **m–p** shows *Cedrela odorata*. Curves were determined using general additive mixed models (*GAMM*) from the gamm4 R package[74]. Crown illumination index values range from 1 for understory trees with no direct light, to 8 for trees with full crown exposure from above and side (see Methods section). Note that in this analysis (in contrast to data shown in Fig. 1) all trees were included even when lower than 1 m in height. Coefficients of determination (R-squared) and significance levels for the regressions are given in Table 2

or even tripling from the youngest to the oldest trees, corresponding to increases in the order of 35–70 ppm over 100 years (Fig. 1a, c, e, g). In contrast, the only coniferous species included in this study, *Pinus*, showed only a very weak trend in $W_i$ of just 4 ppm over 100 years. In all four species the trends in $W_i$ persisted over the full extent of the studied age range, and remained strong after an age of 50 years (Table 1). The only species for which we could test the persistence of trends in $W_i$ for trees older than 100 years was for *Pinus* using sub-fossil data from Helama et al.[39], revealing that even in trees older than 100 years, $W_i$ continued to increase with age (Table 1; Supplementary Fig. 1).

We next compared the observed developmental trends in $W_i$ with time trends in $W_i$ estimated based on traditional tree ring approaches (Fig. 1b, d, f, h). As mentioned these approaches reconstruct historical changes in $W_i$ by looking backwards in time using individual tree ring series of big trees. Thus they implicitly

include the full developmental trajectory for each tree growing from seedling into a large adult tree. Time trends in $W_i$ were collected from the literature (*coloured lines*, see Supplementary Table 1 for details) and from dominant trees from the same sites as the age trends in Fig. 1a, c, e, g (*grey lines*). The comparison reveals that for the three broadleaf species, time trends in $W_i$ derived from dominant trees obtained from literature (*coloured lines*, Fig. 1, see Supplementary Table 1 for details) and complemented with new data collected from the four sites in this study (*grey lines*, Fig. 1) are of a similar or lesser magnitude than the observed change in $W_i$ with tree age. Only for *Pinus* we observe a much stronger increase in $W_i$ over time (i.e., with calendar year) than the increase with biological age (26 vs. 4 ppm (100 yr)$^{-1}$).

**Causes of developmental trends in $W_i$.** To obtain insight into the ultimate causes of the developmental change in $W_i$, we analysed the relationship between the average $W_i$ of tree rings from the

**Table 2 Effects of tree age, diameter, height and crown illumination on intrinsic water use efficiency**

| Species | Age | Diameter | Tree height | Crown illumination | Most parsimonious regression model | Variance explained |
|---|---|---|---|---|---|---|
| *Pinus sylvestris* | 0.005[NS] | 0.053* | 0.001[NS] | 0.131*** | Tree height, light index, diameter | 18% |
| *Quercus robur* | 0.353*** | 0.499*** | 0.476*** | 0.515*** | Tree height, crown illumination | 68% |
| *Fagus sylvatica* | 0.333*** | 0.495*** | 0.394** | 0.541*** | Crown illumination, diameter | 61% |
| *Cedrela odorata* | 0.611*** | 0.662*** | 0.762** | 0.516*** | Tree height, crown illumination | 78% |
| *Linear mixed-effects model (LME) outcome* | | | *t-value* | *P-value* | | |
| *Age* | | | −3.870 | 0.0001 | | |
| *Diameter* | | | 0.427 | 0.5716 | | |
| *Tree height* | | | 7.761 | 0.0000 | | |
| *Crown illumination* | | | 7.331 | 0.0000 | | |

Significance levels and coefficient of determination (R-squared) for each predictor are shown per species. To evaluate which set of predictors best fitted the data for each species, we used Akaike Information Criterion to estimate the most parsimonious regression model[77]. The lower part of the table shows the outcome of linear mixed-effects model for the effects of four predictive variables on $W_i$, including all four species as random effects. Note that the negative effect of tree age on $W_i$ arises from the relationship of age with other predictor variables. Significance levels are indicated as follows: NS not significant; *$p < 0.05$; **$p < 0.01$; ***$p < 0.001$

last 5 years of growth with age, diameter, height and crown illumination. While these factors are intrinsically linked, the comparison (Fig. 2; Table 2) does indicate that ageing per se is not the main cause. For *Pinus, Quercus* and *Fagus*, age explains less variation (lower R-squared) compared to a trees' size (diameter, height) or crown illumination, and in none of the four species is age included in the most parsimonious regression models (Table 2). Variation in $W_i$ is most strongly related to trees' size and crown illumination, which together explain between 56 and 78% of the variation in $W_i$ for the three broadleaf species (Table 2). Similarly, the linear mixed-effects model, including data from all four species, indicates that crown illumination and tree height, and not age, explain the largest proportion of variation in $W_i$ (Table 2). In *Pinus* only a small portion of the total variation in $W_i$ (18%) is explained by any of the measured developmental variables. Finally a comparison of trends in $\delta^{13}C$ with age and height shows that height trends in the three broadleaf species converge nearly to the same magnitude of change in isotope ratios with height (i.e., 1.58–1.77‰ $(10 \text{ m})^{-1}$), while age trends vary more between species (Supplementary Fig. 2).

To assess the contribution of $CO_2$ from soil respiration to the observed developmental effects, we compiled $\delta^{13}C_{air}$ and $CO_2$ data for temperate and tropical forests (Supplementary Fig. 3). The data reveal that during daytime, soil respiration affects $\delta^{13}C_{air}$ and $CO_2$ in the first few metre(s) above the ground with slightly larger effects for tropical forests compared to temperate forests[37, 44]. We estimated the contribution of this effect on observed trends in $W_i$ by taking into account estimated values of $\delta^{13}C_{air}$ and $[CO_2]$ at crown height for each tree (see Methods section and Supplementary Fig. 3). We find that inferred increases in $W_i$ with tree age for *Fagus, Quercus* and *Cedrela* are rsp. 11, 14 and 20% lower when using below-canopy values of $CO_2$ and $\delta^{13}C_{air}$ (Supplementary Fig. 4). However, an alternative analysis in which we excluded data from small trees (e.g., lower than 1, 2, 5 or 10 m in total tree height) demonstrates that $W_i$ trends with tree height remain strong, even for trees bigger than 5 or 10 m in height (Table 1). Together these results suggest that below-canopy variation in source $CO_2$ explains only a very small portion of variation in $W_i$.

## Discussion

We show here that $W_i$ increases strongly with tree age in the three broadleaf species *Quercus, Fagus* and *Cedrela* and weakly in *Pinus*. This is an important finding as it invalidates the assumption of most tree ring isotope studies aiming at reconstruction of tree water use efficiency over time that developmental

trends are negligible. Using a simple, size-stratified sampling approach we demonstrate that developmental increases in $W_i$ in individual tree ring series are strong for three out of four species, and thus will be wrongly interpreted as responses of trees to global change unless the records are corrected for these effects (Fig. 1). For example, comparison of the increases in $W_i$ with age for *Cedrela* from our study with results from Nock et al.[16] and Hietz et al.[45] shows that observed increases in their tropical tree species (including *Cedrela*) are of the same order of magnitude as the developmental effects for *Cedrela*. The lack of size-stratified sampling in these studies makes it hard to unambiguously isolate the $CO_2$ effects. Our findings similarly question to what degree reported increases in $W_i$ in European *Quercus* and *Fagus*[10, 11, 41, 46] are indeed due to $CO_2$, or rather primarily the result of developmental effects. We find that *P. sylvestris* is much less affected by tree development, consistent with other indications for this species[42, 43]. Reported estimates of $W_i$ increases for this species in, for example, the European tree ring isotope networks[11, 41] are thus more likely to be correctly attributed to external changes like, e.g., rising $CO_2$ although still with a caveat since the strength of developmental effects may vary between sites and with local climate.

A second important finding of our study is that developmental trends are not limited to the earliest phases of a trees' life, but within the maximum age limits of our sample seem to last over the entire lifetime of trees. In *P. sylvestris*, $W_i$ increases with age even for trees older than 100 years (Supplementary Fig. 1), reflecting earlier observations by Helama et al.[39] using $\delta^{13}C$. For the three broadleaf species $W_i$ also continues with age, even after trees have reached an age of 50 years (Table 1). The age of 50 years is a commonly suggested cut-off to remove developmental effects (in this connection usually dubbed 'juvenile effect')[24, 43, 46, 47]. Our results indicate this is not a valid approach as $W_i$ continues to increase even in old trees.

Analysis of the relationship of $W_i$ with various tree developmental characteristics (Fig. 2; Table 2) shows that the observed trends are not driven by age per se. Instead increases in tree height and changing crown illumination over a trees' life are the principal drivers. In our analysis these two variables, tree height and crown illumination, explain over 60% of the variation in $W_i$ in the broadleaf species. As tree height and crown illumination increase simultaneously in trees over the course of their life, it is however impossible to fully separate the effect of the two variables with our methods. Nevertheless, for *Fagus* and *Quercus*, we collected and analysed samples from trees that were of similar size (all small saplings), but which differed strongly in their light environment. $W_i$'s differed markedly revealing light exposure as an important

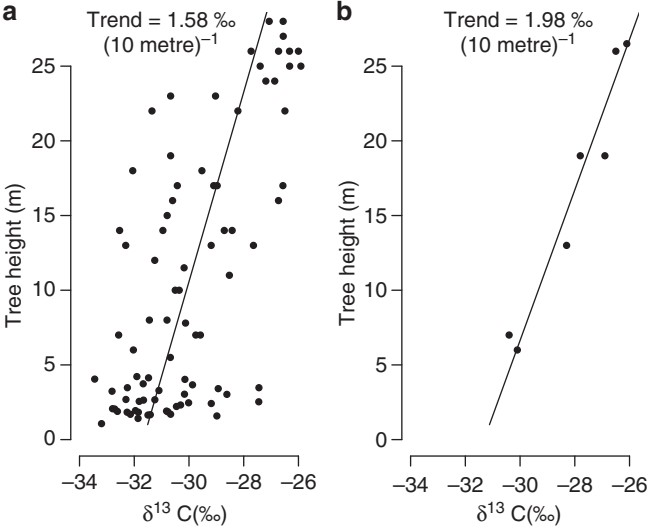

**Fig. 3** Comparison of height- $\delta^{13}$C curves derived from tree ring and leaf $\delta^{13}$C -data for *Fagus*. Data in **a** are from this study (i.e., bulk samples of the last five rings from different sized trees). Leaf data in **b** are from Schleser[56], collected at different heights in the canopy from a single *Fagus* tree in Germany

driver of variation in $W_i$, irrespective of the trees' size. For *Cedrela* tree height plays a more important role compared to crown illumination, likely via hydraulic limitation, while the combined results for all four species suggest that variation in $W_i$ is driven by these two variables approximately equally.

The effect of tree height on leaf isotope discrimination is well-known and attributed to increasing constraints on water transport to the canopy with increasing tree height[23]. Irradiance has an additive effect on the height–discrimination relationship by increasing assimilation more than stomatal conductance for sunlit leaves resulting in lower partial pressure of $CO_2$ inside the leaf, at any given height[23, 48]. While these two factors seem to be the main controls of change in isotope discrimination with tree height, various related plant physiological changes may play a role as well. For example, change in rooting depth and water uptake[49], leaf morphology and physiology[28, 32, 33, 50], leaf nitrogen through its effect on photosynthetic capacity[51], changing leaf area to sapwood area ratios[34, 35] and increases in average branch length[52] as trees grow bigger are all known to affect isotope discrimination. In addition, decreases in relative humidity from lower to upper canopy may also affect isotope discrimination when trees grow higher[53]. The contribution of soil-respired carbon to the tree height-$\delta^{13}$C relationship, which has been the focus of several of the earlier isotope studies[26, 36], seems to be comparably small. In the case of *Cedrela* growing in dense tropical rainforests, where these soil contributions were found to be highest[37, 44], we estimated that no more than 20% of the trend in $W_i$ can be explained by soil respiration, with soil respiration contributions to $W_i$ trends in the two temperate broadleaf species being even lower (11 and 14%, Supplementary Fig. 4). We further find that for all four species, $W_i$ continues to increase with tree height beyond the first few metres (Table 1). Thus, changes in $W_i$ with tree height observed in this study are mainly related to plant physiological and environmental changes, such as tree hydraulics and crown illumination.

Our analysis includes only four species, and thus caution is needed with generalisation to other species and biomes. Nonetheless while the type of tree ring studies that we performed here are scarce, various studies have analysed variation in leaf isotope discrimination with tree height. This raises the question

whether variation in isotope discrimination with leaf height in the canopy can be used as an indicator for trends in wood $\delta^{13}$C (i.e., for the change in tree ring $\delta^{13}$C with total height of the tree at the time of ring formation). The answer is not so clear because wood $\delta^{13}$C integrates the isotope signal from the entire canopy, mixing carbohydrates produced by leaves at different height[54]. For trees with deep crowns and heavy self-shading one would therefore expect trends in wood $\delta^{13}$C with tree height to be weaker than the trends of leaves with height[23, 55]. On the other hand, sunlit leaves at the top of the canopy are expected to assimilate comparably more carbon and contribute disproportionally to the average isotope signal found in the main trunk[54]. A review of published isotope data by McDowell et al.[23] shows indeed weaker trends with tree height for wood compared to leaves. Notwithstanding these results, we observe that the average change in carbon isotope discrimination in leaves with tree height (1.9 ‰ $(10\,m)^{-1}$ from the analysis of McDowell et al.[23] is quite similar to the trends in carbon isotope discrimination in wood for the three broadleaf species in our study, which vary between 1.6 and 1.8 ‰ $(10\,m)^{-1}$ (Supplementary Fig. 3). Furthermore, a comparison of height trends in tree ring $\delta^{13}$C in our study with leaf $\delta^{13}$C collected at different heights in a single *Fagus* tree by Schleser[56] shows that trends in wood $\delta^{13}$C are only slightly weaker compared to foliar $\delta^{13}$C trends (Fig. 3). This suggests that leaf $\delta^{13}$C may indeed be used as an indicator for the existence of trends in wood $\delta^{13}$C, although not necessarily for the absolute magnitude of the trends. According to the meta-analysis by McDowell et al.[23] height trends in leaf isotope discrimination are common across a wide range of habitats and species (37 of the 38 different species showed a decrease in leaf discrimination with tree height, or an increase in $W_i$). While more investigation into wood and leaf $\delta^{13}$C trends with height for a large range of species is needed, this is a strong indication that height trends in tree ring $\delta^{13}$C are probably very common and may be the rule rather than the exception.

Surprisingly, for both *Fagus* and *Quercus*, we find that time trends in $W_i$ for the dominant trees (by looking backwards in time using individual tree ring series of big trees, Fig. 1b, d, f, h) are much weaker than the increase in $W_i$ with tree age for the same species from the same sites (Fig. 1a, c, e, g). We attribute this to distinctly different growing conditions for these dominant trees when they were small, compared to current small trees from our sample. The dominant *Fagus* and *Quercus* trees from Bishop Wood, UK, were indeed planted and probably grew up in quite open conditions with high light availability and low humidity. This is in agreement with our results about the overwhelming influence of historical growing conditions on trees' variation in $W_i$. It also calls for more investigation into the influence of historical stand development, specifically the role of competition, light availability and height gains on $W_i$ trends.

Overall, our results reveal a clear need to account for developmental effects in tree ring $\delta^{13}$C if the aim is to extract long-term trends in $W_i$ due to, for example, $CO_2$ or climate. This recommendation has been made in previous studies[25, 40, 42], but has rarely been followed. Recent tree ring isotope studies from the European isotope networks do not account for developmental effects, and rings of different ages for the same calendar year are even commonly pooled prior to $\delta^{13}$C analysis[11, 41]. Such approaches will inevitably obscure developmental trends, and make it virtually impossible to separate such trends from real changes in $W_i$. Developmental effects in tree ring data are commonly removed using Regional Curve Standardisation[57–59] applied to age. Our results argue instead for an approach that removes the effect of tree size (similar as in van der Sleen et al.[8]), as this is the parameter most strongly related to $W_i$. It should be noted however that all of these approaches are prone to additional biases[60, 61]. These biases arise because of differences in survival

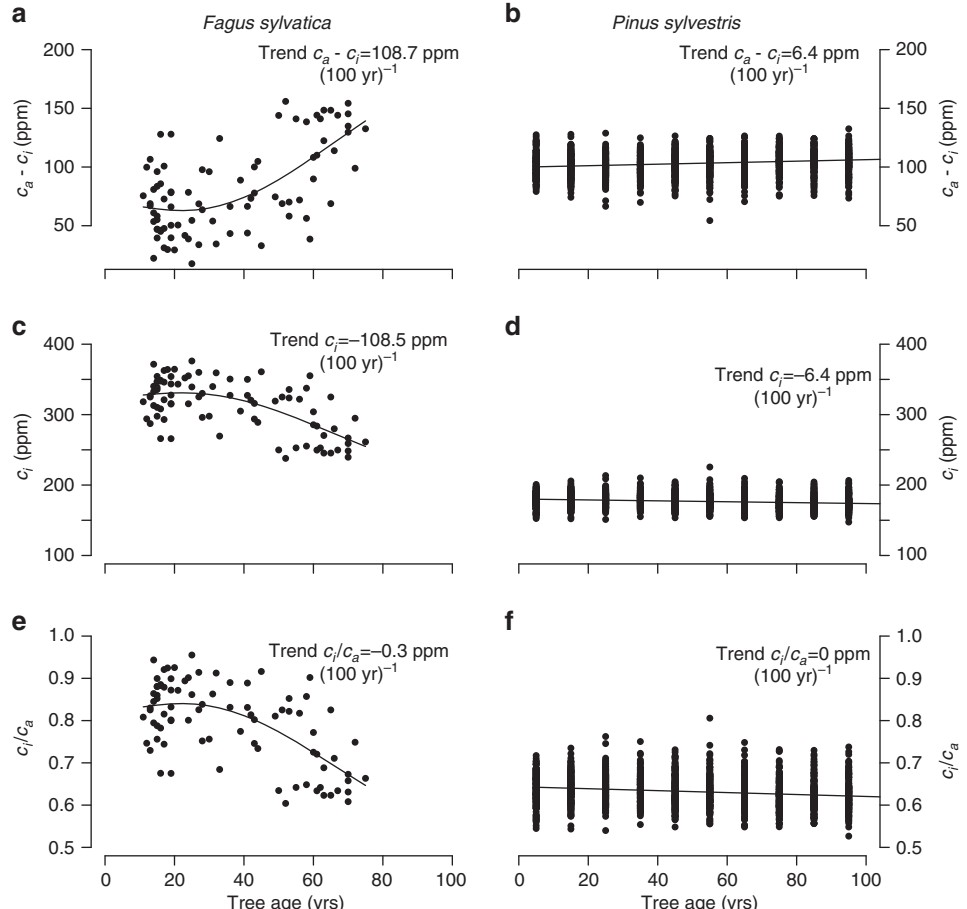

**Fig. 4** Developmental trends in gas-exchange regulation diagnostics for *Fagus* and *Pinus*. **a**, **b** show air-to-leaf $CO_2$ difference ($c_a−c_i$), **c**, **d** show leaf internal $CO_2$ ($c_i$) and **e**, **f** show the ratio of leaf-to-air $CO_2$ ($c_i/c_a$) plotted against tree age. These values are calculated from the average $\delta^{13}C$ (see Methods section for formulas) of the five outermost rings across a range of trees differing in tree age (for *Fagus*) or from sub-fossil trees (for *Pinus*, Helama et al.[39]). In both cases, age trends were controlled for changing $CO_2$ levels. **a**, **c** and **e** show results for *Fagus* and **b**, **d** and **f** for *Pinus*

changes for fast- and slow-growing trees (which may have different $W_i$), and/or due to specific stand history development. Bias correction as described in Brienen et al.[60] is needed when these approaches are used to estimate long-term $W_i$ changes in response to $CO_2$ and climate. The importance of knowing specific stand history of the sampled trees is nicely illustrated in the comparison of developmental and time trends for *Quercus* and *Fagus* (Fig. 1); using the age-$W_i$ curve in these two species to detrend the historical time series of $W_i$ in current dominant trees would have resulted in negative trends in $W_i$ over time. The very site-specific nature of these effects means that it may be very difficult or perhaps impossible to remove these effects from many previously published records, but it is recommended that future studies collect and evaluate stand history data.

It is well-known that rising $CO_2$ affects stomatal regulation of leaf gas exchange[5]. As a general framework to evaluate trees' responses to increases in $CO_2$, researchers have proposed three strategies for homoeostatic gas-exchange regulation[11, 12, 17, 25], which are maintaining a constant leaf internal $CO_2$, $c_i$, a constant air-to-leaf $CO_2$ difference, $c_a−c_i$, or equivalently $W_i$ ($\equiv (c_a−c_i)/1.6 \equiv A/g_w$) or a constant ratio of leaf-to-air $CO_2$, $c_i/c_a$. These strategies have guided the interpretation of tree ring and isotope-derived changes in leaf gas exchange to increasing $CO_2$. However it is largely unknown to what extent trees follow just one of these specific gas-exchange strategies over their lifetime, if any at all, and how they vary between species. This

question is relevant for the interpretation of long-term responses to $CO_2$, and is also interesting by itself. In Fig. 4, we show age trends in $c_a−c_i$, $c_i$ and $c_i/c_a$. for *Fagus* and *Pinus*, as derived from the developmental trends in carbon isotope discrimination (see Methods section). Trees clearly do not adhere to a single gas regulation strategy over their life. For example, in *Fagus* observed increases in $c_a−c_i$ correspond to strong decreases in $c_i$ and decreases in $c_i/c_a$. with tree age. Thus, young Fagus trees have significantly higher leaf internal $CO_2$ concentrations, $c_i$, compared to older, and thus larger, trees, presumably due to increased stomatal limitations on assimilation as trees grow taller. There are also strong differences between species, as *Pinus* shows much weaker change in $c_i$ and $c_i/c_a$ with age. These differences may be related to taxonomy (conifers vs. broadleaf species), difference in shade tolerance between species or growing conditions (open sites vs. forests). These results illustrate that trees do not adhere to a single gas regulation strategy over their lifetime, but that strategies vary between different developmental stages and species, reminiscent of observations of gas-exchange responses to $CO_2$[12].

Finally, we would like to emphasise that, while our results cast doubt on interpretations of tree rings isotope trends of $W_i$ in the context of long-term responses to $CO_2$, the purpose of this paper is not to dispute the existence of beneficial $CO_2$ effects on plant growth. As already mentioned, various studies that do control for developmental effects in tree ring data report relatively large increases in $W_i$[8, 40, 62] (but see Marshall and Monserud[38]),

consistent with ecosystem flux measurements[7], and with $\delta^{13}C$ observations in paleo records and $CO_2$ enrichment studies[12]. However, tree ring $\delta^{13}C$ can only be used for improving our understanding of $CO_2$ effects on $W_i$ if developmental effects are taken fully into consideration. We also would like to point out that the identified developmental effects have similar implications for the use of tree ring $\delta^{13}C$ to study tree responses to, e.g., nitrogen deposition[14], air pollution[63] and climate[11], as well as, for any palaeo-climate reconstructions[24, 64].

In summary, we here report that some species show very strong developmental changes in isotope discrimination with a doubling or even tripling of $W_i$ over a trees' lifetime without any change in atmospheric $CO_2$. These effects are highly species-specific and predominantly related to increases in tree height and accompanying increases in light availability. In three of the four species the effects of developmental change in $W_i$ were of similar magnitude as estimated time trends of $W_i$ from tree ring $\delta^{13}C$ series of big trees, suggesting that these may very well just be artefacts. We also observed that the developmental trends are not limited to the earliest life phases, but continue all the way through to old age. These results provide a stark warning against ignoring developmental trends and show that existing reports on increases in $W_i$ need to be interpreted very cautiously, and effectively require re-evaluation.

## Methods

**Study species and field sampling.** In this study, we focus on four different species, *P. sylvestris* (Scots pine), *Q. robur* (pedunculate oak), *F. sylvatica* (common beech) and *C. odorata* (Spanish cedar) for which we present new and original data and compiled literature data (Supplementary Table 1). *Pinus*, *Quercus* and *Fagus* were chosen as these have been extensively used in European dendro-isotope studies[11, 13]. *Cedrela* was included as a representative for tropical trees, which are increasingly being used in tree ring isotope studies[8, 16, 45]. Samples for *P. sylvestris* (100 trees) were collected in April 2015 from a natural Pine stand in the Cairngorms National Park, Scotland near Loch-an-Eilein (57.13N, −3.83E). Most of the sampled Pine trees were growing in full sunlight, but we also sampled individuals from a young dense stand with high levels of competition. For *P. sylvestris* we also present sub-fossil isotope records from 180 ancient trunks, which dated between 5630 BC to 1930 AD, and were recovered from lake sediments from Finnish Lapland (for details see Helama et al.[39]). Samples of *Q. robur* (78 trees) and *F. sylvatica* (81 trees) were collected in 2015 from Bishop Wood, which is a mixed evergreen broadleaf forest in North Yorkshire, UK (53.79N, −1.15E) managed by the UK Forestry Commission. The oldest *Quercus* and *Fagus* trees in this forest were planted in 1922. For *Fagus* we collected additional 33 samples from small trees from a park landscape in Leeds, UK (53.82N, −1.58E) to complement our sample with saplings from across a range of different light environments (i.e., full sunlight to deep shade). *C. odorata* samples (75 trees) were collected from two different sites of natural lowland tropical moist forests located in the province of Pando in northern Bolivia in October 2002 in Purissima (−11.40N, −68.72E), and in 2011 in Selva Negra (−10.08N, −66.30E). For all sampled species, the strategy consisted of collecting cores or stem sections from across the full size range of the existing populations, assuring an even sample distribution across size classes. We collected increment cores from trees >5 cm in diameter, but for saplings and seedlings with diameters that were too small to core, we cut full stem sections. In the case of *Cedrela*, we also collected additional large discs or disc sections from trees that were felled for their timber (Brienen and Zuidema[65]). The tree cores were taken using a 5 or 10 mm increment borer, and generally at 1.3 m above the forest floor when trees were large, or at somewhat lower heights of 30–50 cm above the forest floor for smaller trees. Complete stem sections for small saplings and seedlings shorter than 2.5 m in height were generally taken at a height between 5 and 15 cm above the forest floor. Increment cores were taken from at least two radii for *Quercus*, *Fagus* and *Pinus*, and generally in three direction for *Cedrela*. Tree ring analysis on discs for *Cedrela* always made use of at least three radii. For all trees, we recorded the diameter at breast height, estimated the total tree height either by eye (in the case of *Cedrela*) or the use of a Nikon Pro Laser Rangefinder, and assessed light availability using the modified crown illumination index (CII) of Clark and Clark[66]. We used the following CIIs classes: 1 = no direct lateral or overhead light; 2a = little direct lateral light, no overhead light; 2b = some direct lateral light, no overhead light; 2c = substantial direct lateral light, no overhead light; 3a = little direct overhead light; 3b = substantial direct overhead light; 4 = more than 90% of crown receives full overhead direct light; and 5 = full overhead and lateral, direct light. To perform tree ring analysis, we glued cores to wooden bases, and prepared surfaces by sanding or using a core microtome. All four species form clear and annual rings, including tropical *Cedrela* in the two sites used here[67], and tree ages were estimated by ring counting without crossdating. For cores with

missing piths we calculated the distance to the pith and used mean growth rates for that species and diameter class to estimate the number of missing rings.

**Isotope analysis.** We isolated and analysed bulk samples containing the last five rings for *Fagus*, *Quercus* and *Pinus*. For *Cedrela*, we analysed the isotope ratio for each ring over the last 5 years, and then took the average of those years. For *Cedrela* the sampling date differed by 9 years between the two sites that were used, but this did not affect the results, as we find no difference in $W_i$ between these periods (data not shown). Long isotopes series from large, dominant trees were reconstructed either by measuring $\delta^{13}C$ for each individual ring (*Cedrela*), or by measuring $\delta^{13}C$ in ring sections of 10 years (*Fagus*, *Quercus* and *Pinus*). Rings or ring sections were cut up using a scalpel, and cellulose was extracted following the batch method of Wieloch et al.[68]. Cellulose was homogenised using a mixer mill (Resch MM301) and then freeze-dried. Samples were then weighed into tin capsules for isotope analysis. The isotope analysis was done at four different labs: the British Geological Survey's Stable isotope Facility (Part of NERC Isotope Geosciences Facilities) (NIGF, Keyworth, Nottingham, UK), the School of Earth and Environment (SEE) at the University of Leeds, the German Research Centre for Geosciences (GFZ, Postdam and Julich, Germany) and the Laboratory of Chronology at the University of Helsinki (Finland). Analysis of the *Cedrela* samples from Selva Negra were performed at British Geological Survey utilising a Costech Elemental Analyser (EA) online to a VG TripleTrap and Optima dual-inlet isotope mass spectrometer (IRMS). *Cedrela* cellulose samples from Purissima at GFZ were converted to $CO_2$ in an excess of oxygen using an elemental analyser (Carlo Erba NA 1500) coupled to an OPTIMA (Micromass Ltd, UK) IRMS. The remaining samples for *Fagus*, *Quercus* and *Pinus* (Scotland) were analysed at University of Leeds using an Elementar Vario Pyrocube coupled to a GV Isoprime mass spectrometer. Analysis of the sub-fossil samples from Finland was analysed in duplicate on a DELTA Advantage isotope ratio spectrometer coupled to a CN2500 elemental analyser at the Laboratory of Chronology at the University of Helsinki (Finland). All laboratories used standards that included IAEA-CH7 and/or IAEA-CH3, or in-house standards calibrated against at least one of these IAEA international standards.

**Data analysis.** Carbon isotope ratios ($\delta^{13}C$) were calculated as

$$\delta^{13}C_{plant}[‰ \text{ vs. } V-PDB] = (R_{sample}/R_{standard} - 1) \times 1000 \qquad (1)$$

with $R$ representing the abundance ratios, $^{13}C/^{12}C$, of the sample and the standard (Vienna Pee Dee Belemnite (V-PDB)). The unit permille plant carbon isotope ratios were converted to plant to air isotope discrimination ($\Delta$, Farquhar and Richards[69]),

$$\Delta^{13}C_{plant} = (\delta^{13}C_a - \delta^{13}C_{plant})/(1 + \delta^{13}C_{plant}/1000) \qquad (2)$$

with $\delta^{13}C_a$ representing the isotopic composition of atmospheric $CO_2$, which is becoming depleted in heavier $^{13}CO_2$ over the last two centuries due to combustion of isotopically light fossil fuels. We used records of $\delta^{13}C_a$ obtained from Antarctic ice cores[70], complemented with recent data from Mauna Loa from http://www.esrl.noaa.gov/gmd/ccgg/trends/full.html. Following Farquhar et al.[71], plant discrimination is assumed to be related to the ratio of intercellular to atmospheric [$CO_2$] ($c_i/c_a$) by the following equation:

$$\Delta^{13}C_{plant} = a \times \left(\frac{c_a - c_i}{c_a}\right) + b \times \left(\frac{c_i}{c_a},\right) \qquad (3)$$

where $a$ (=4.4‰) results from the slower diffusion of $^{13}CO_2$ relative to $^{12}CO_2$ through the stomata and $b$ (=27‰) is the fractionation by Rubisco against $^{13}CO_2$ inside the leaf. $C_i$ can be calculated directly for each ring, or ring section using $c_a$, atmospheric $CO_2$ data from http://www.esrl.noaa.gov/gmd/ccgg/trends/full.html. This equation ignores mesophyll conductance[72] and assumes that post-photosynthetic factors beyond the formation of primary leaf sugars[73], do not change discrimination. It is well-known that this assumption is not correct, as various environmental and physiological processes (including temperature and tree- or leaf age) affect mesophyll conductance and post-photosynthetic discrimination[72, 74]. However, considering current lack of knowledge on these processes[73], and specifically how they vary between species and environmental conditions, we choose not to correct for the mesophyll conductance term or leaf-to-wood offsets. Many studies on water use efficiency similarly ignore these effects, or apply a fixed term to account for post-photosynthetic processes[11], which does not change the trend analysis which is the focus of our analysis.

Intrinsic water-use efficiency ($W_i$) is defined as the ratio of assimilation rate ($A$) to stomatal conductance for water vapour ($g_w$), and can be calculated if $c_i$ and $c_a$ are known using Fick's Law,

$$A = g_c \times (c_a - c_i) \qquad (4)$$

Stomatal conductance for water ($g_w$) is $1.6 \times g_c$ (stomatal conductance for $CO_2$), where 1.6 is the ratio of molecular diffusivity of water vapour and $CO_2$ in air.

Combining Eqs. [2], [3] and [4], we calculate $W_i$ as follows,

$$W_i \equiv A/g_w = A/(1.6 \times g_c) = (c_a - c_i)/1.6 = c_a \left(b - \Delta^{13}C_{plant}\right)/1.6(b-a) \quad (5)$$

Note that we use the shorter parts per million (ppm) notation as units for $W_i$, which is the equivalent of the mole fraction ($\mu$mol mol$^{-1}$). The sub-fossil $\delta^{13}C$ data from *Pinus* from Helama et al.[39] originated between 5630 BC and 1930 AD, and to calculate $W_i$ for these trees we assumed a constant, pre-industrial atmospheric $CO_2$ level of 280 ppm and $\delta^{13}C_{air}$ of $-6.4\permil$ based on Francey et al.[70]. Slight temporal variations in $CO_2$ and $\delta^{13}C_{air}$ during this period will not cause any systematic bias in the calculation of development trends in $W_i$ as the trees represent an average from very different time periods (Helama et al.[39]).

To study the effect of tree age, diameter, height and light availability on carbon isotope discrimination, we plotted $W_i$ against each of these variable for rings that were formed in the same calendar years thus controlling for the effect of $CO_2$ on $W_i$. We estimated developmental trends and time trends in $W_i$ using general additive mixed models from the gamm4 R package[75]. To estimate long-term linear $W_i$ trends we used linear mixed-effects models from lme4 R package[76]. In the comparison of developmental trends with time trend in tree rings, we excluded trees lower than 1 m in height from the developmental trend analysis (Fig. 1a, c, e, g) to avoid influence of soil-respired carbon, and because tree ring analysis usually does not sample trees at height lower than 1.30 m. Our estimates of time trends in $W_i$ included time trends from trees collected in this study complemented with literature data. To account for differences in sampling intensity between studies, we used weighting in the calculations of the time trends according to the number of study sites for each $W_i$ trend. See Supplementary Table 1 for a list of sourced literature data and the number of sites that were included. For display purposes (Fig. 1) we smoothed the annual *Cedrela* $W_i$ series for each tree using a cubic smoothing spline function.

**Soil respiration**. To estimate the contribution of soil respiration to age and height trends in tree ring-derived estimates of $W_i$, we compiled literature data on differences in $CO_2$ concentrations and $\delta^{13}C$ in $CO_2$ ($\delta^{13}C_{air}$) under the forest canopy compared to values above the canopy. We approximated the change in $CO_2$ and $\delta^{13}C_{air}$ with height above the forest floor by a negative exponential function for temperate and tropical forests (Supplementary Fig. 4). Using total tree height data for individual trees collected in our study, we then estimated the available $CO_2$ concentration and $\delta^{13}C_{air}$ for each tree. As total tree height is probably not the best measure of a trees' actual uptake of $CO_2$, we assumed that average $CO_2$ uptake by the trees' canopy occurred slightly below the top of the trees, at $0.9 \times$ total tree height. We then re-calculated $W_i$ for each tree corrected for below-canopy effects, and re-estimated $W_i$ trends with age and height. This analysis was done only for the three broadleaf trees, and not for *Pinus* as this species grew mostly in open areas.

**Data availability**. All the data used in this publication are available from the authors upon request.

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

## Acknowledgements

We thank Thijs Pons, Jessica Baker, Sophie Fauset and three reviewers for comments on previous draft of the manuscript, Jessica Baker and Bruno Ladvocat for help with fieldwork, and Rob Wilson for logistical support with the Scottish Pine trees. Chris Kendrick performed the isotope measurements at the British Geological Survey. We acknowledge NOAA-ESRL for making available atmospheric CO₂ records. This work has been supported by the National Environmental Research Council (UK) through a NERC Research Fellowship (grant NE/L0211160/1), NERC standard grant (NE/K01353X/1) and by NERC Isotope Geosciences Facilities grants (IP-1424-0514 and IP-1314-0512).

## Author contributions

R.J.W.B., E.G., S.C. and R.N. conceived the study, S.C., R.N., S.B., G.H., M.J.L. and T.H. performed and/or oversaw the isotope measurements for this study, R.J.W.B., S.C., M.C., L.A., M.T., K.M., M.O. collected field samples and performed standard dendrochronological analysis, R.B. and E.G. wrote the paper and all authors provided comments on the manuscript.

## Additional information

**Competing interests:** The authors declare no competing financial interests.

