## [Peer Review File · Nature Communications]

Reviewers' Comments:

Reviewer #1 (Remarks to the Author)

In this paper, the authors address the research question of whether increasing trends in water use efficiency, measured using carbon isotopes ratio in tree-ring series, are truly due to increasing gained carbon per unit water lost or need to be ascribed also to other variables.

In order to solve this question, the authors analyse developmental trends in tree ring carbon isotopes in four species (three broadleaf and one conifer) sampled in various sites and years. They also analyse data available in literature.

The raised question is scientifically sound and of wide scope for the applications in climate reconstructions through tree-ring series.

The different sections of the paper need improvement. Some concerns about statistics and some major weaknesses are listed below.

1. Lines 49-51. Do these studies analyse 1 core per 1 tree to reconstruct carbon isotopes trends in tree-ring series?
2. Lines 65-68. Rooting depth should be also considered (also at lines 239-241).
3. Lines 74-75. This is the hypothesis, I suppose. The way this concept is written, makes it seem such inconclusiveness has been already proved.
4. Lines 98-100. Aim n. 3 is not clear. Moreover, it seems it has not been addressed in the paper.
5. Lines 127-130. Not clear. This concept needs to be explained more in detail.
6. Lines 142-144. This statement is not fully supported by Fig. 2 and Table 2. Most analysed predictive variables have $p < 0.001$ level of significance. See for example the value of R-squared for the predictive variables "diameter" and "crown illumination": is the difference between 0.499 and 0.515 enough to state that crown illumination has a stronger effect than diameter? Probably, from a statistical viewpoint, it can be correct, but what about the biological meaning? Moreover, a value of R-squared equal to 0.131 is considered very highly significant (even more, a value of 0.053 is significant too): this is probably due to a very high number of observations, but I believe values so close to zero mean no relation at all but only a statistical trick (this also applies to table 1). Moreover, the same information should not be reported both in figure and table.
7. In general, it is not clear which information is derived from data gained from this experiment and which from meta-data.
8. The discussion is too long, sometimes speculative (e.g. lines 270-281) and some parts seem a sort of rephrasing of the introduction. It should be reduced in length and better focused on results. Apart from stating which are the bias in published papers, some solutions should be also suggested.
9. Lines 144-150. This is not straightforward. Please, check.
10. Methods are somewhat confusing. Are all data presented derived from previously published papers? Please refer also to comment n. 7. A summary information about the study sites, climate, plant age and features of the stands should be added. Were the tree rings dated only visually, without any cross-dating?
11. Line 387. "When possible" is not acceptable. Please, quantify.
12. The authors report that isotope analyses were done on the last 5 rings/years. Since the plants were not all collected in the same year (lines 365-394), does this mean that tree rings from different calendar years were analysed and compared? The paragraph from line 408 is confusing. Please explain better the sampling.
13. It is not clear to what extent the other environmental drivers and intrinsic plant traits (diameter, height and crown illumination) affect estimation in water use efficiency.

Reviewer #2 (Remarks to the Author)

Tree-ring stable carbon isotopes have been used to infer CO₂ effects on intrinsic

water-use efficiency of growing trees. This manuscript claims that such studies should account for confounding due to tree age and height, which increase in parallel with the recent rise in atmospheric CO₂. The argument has been made before, but it is still frequently ignored, even in some high-profile, recent papers. This issue was brought into high relief by Keenan et al.'s nature paper in 2013, which used eddy flux to infer extreme increases in water-use efficiency in recent decades, dismissing the tree-ring isotope data, which shows smaller effects, as ridden with artifacts. This paper describes one of those artifacts and provides a means of correcting for it. Perhaps this will be seen in the future as a step toward reconciliation of tree-ring and eddy-covariance techniques.

Specific comments: Change title: "Developmental trends in tree-ring carbon isotopes affect estimates of water-use efficiency responses to CO₂"

Lines 60 to 69: I appreciate this summary of "developmental effects," but I would present them differently. I suggest presenting the height effect first because it should be universal, controlled by the gravitational effect on water potential. In contrast, many trees do not grow under the canopies of other trees, but rather grow in even-aged stands. The beech studied here does grow under canopies, but the pine studied here does not. The light effects might thus be expected in the beech, but not the pine. Similar effects might be expected in the refixation of respired CO₂, which is probably more significant for trees growing up under the canopy than for short trees growing up after a disturbance. Refixation had minimal effects in an even-aged conifer stand with an open canopy (Brooks et al., 1995), which is again more similar to what one would expect in the Finnish pine trees studied here.

Note also that after discussing these three effects, much of the results section deals with "age" as a predictor variable. I understand that age can be measured more precisely, but I advise that the section describing alternative causes should be concluded by a statement that age is what you could actually measure.

Line 70: "very few" studies have looked at this before? It seems to me that this gives insufficient credit to previous work, some of which laid out these issues in detail.

Line 118, Fig. 1, and elsewhere: the authors express iWUE in per mil when it should be umol/mol or ppm. This is confusing because the units of isotopic composition are per mil. This must be corrected.

Line 226-229: not convinced you can use the light environment of similar-sized trees to infer the past light environment of mature trees. Oak and especially beech trees can grow in either bright light or heavy shade. At least this approach should be better justified and perhaps it should be removed from the manuscript.

line 237: this was demonstrated well before reference 25. see Ehleringer et al. in 1986.

line 245: as they say this effect is rather small. Why emphasize it in the introduction?

Line 253: An alternative is to present these species as examples of shade tolerant (beech), intermediately tolerant (oak), and intolerant (pine) species. This would add to the generality and show a better connection to the literature.

line 265: missing reference

line 340: note that this comment agrees with Voelcker et al.

line 345: but not reference 42, which shows constant Wi

Reviewer #3 (Remarks to the Author)

I found the manuscript interesting in sorting out the effects of age, height and exposure, although I had thought some of these issues had been dealt with earlier. Given that this work is to be applied to analysis of the effects of eCO₂ and climate change, I found it surprising that there is no ref to the work of Schubert, who applied a unifying relationship between carbon isotope fractionation and pCO₂ within C₃ land plant tissues towards interpreting terrestrial organic sediments in the fossil record (Schubert and Jahren, 2013, 2015; Cui and Schubert, 2016). This work led to ancillary studies designed to evaluate the confounding effects of pCO₂ and climate on tree-ring δ¹³C values (Schubert and Timmermann, 2015; Trahan and Schubert, 2016).

Intrinsic water use efficiency is introduced without real explanation. One has to wait for the Methods. But once there I could not understand the result. But just getting to this point is frustrating – wondering what is meant by line 91 (“lifetime patterns of Wi”).

Line 118 How does Wi have the units of per mil? If this the trend in ¹³C it is huge. Or does it just mean 3.5 to 7 %?

Line 237 Height effects: see

Givnish TJ, Wong SC, Stuart-Williams H, Holloway-Phillips M, Farquhar GD. 2014. Determinants of maximum tree height in Eucalyptus species along a rainfall gradient in Victoria, Australia. *Ecology* 95(11), 2991–3007

Line 265 “Review of published isotope data by ref. shows indeed weaker trends with tree height for wood compared to leaves.” Is there a missing reference here?

Line 441 Eq 2. There is an unnecessary (and unmatched) bracket here. The reference here should be Farquhar and Richards 1984.

Line 452 Eq3. This is Farquhar et al. 1982.

I think I also derived Eq5. But that is not my main worry.

Line 480 “Note that we use the shorter permille notation (‰) as units for Wi which is the equivalent of the molar fraction (μmol mol⁻¹).”

Firstly the CO₂ concentrations are in mole fractions, not molar fractions. Secondly I was unsure at first what you meant by “the equivalent”. But re-reading it and the results given elsewhere in the manuscript it seems to me that you have equated the CO₂ gradient in μmol mol⁻¹ (parts per million) to something in per mil, which is parts per thousand. In other words unless there is a definition of Wi that is new (and therefore should have been introduced much earlier in the text), the results are out by a factor of 1000.

Line 57 You have equated Wi with Ca-Ci. Shouldn't it be (Ca-Ci)/1.6?
Again the units cannot be parts per thousand.

Response to reviewers.

We would like to thank reviewers for their suggestions and questions, which has allowed us to improve the manuscript. Below we respond to all of the reviewers' comments and explain for each comment the changes made in the manuscript.

Due to comments on the methods and flow of the manuscript, we have tried to clarify our methods and improve the overall flow in the following ways;

- We rephrased parts of the method section and added an overview of the data were collected in this study to the SI Table 1. We also corrected an error in the units for water use efficiency, which caused some understandable confusion for the reviewers.
- We have improved the overall flow of the manuscript by cutting out some repetition between the introduction and discussion, and rephrasing parts that were not so clear. This led to a slight shortening of the discussion.

Along with a new version of the manuscript, we also submit a version highlighting the changes that we made in the manuscript using track changes.

Specific response to reviewers' comments in blue below.

Reviewer #1 (Remarks to the Author):

The different sections of the paper need improvement. Some concerns about statistics and some major weaknesses are listed below.

We have highlighted the changes made in the manuscript to improve the overall flow and clarity of the different sections.

1. Lines 49-51. Do these studies analyse 1 core per 1 tree to reconstruct carbon isotopes trends in tree-ring series?

As a general rule, these studies analyze carbon isotope trends from just one core (or one radial section), but some “pool” the wood from different trees for the same calendar year prior to analysis (see Saurer et al 2014, Frank et al. 2015).

2. Lines 65-68. Rooting depth should be also considered (also at lines 239-241).

Yes, we agree that variation in rooting depth and water uptake between small and large trees may also affect the isotope signal. We have added this to the discussion along with a reference to Dawson 1996.

3. Lines 74-75. This is the hypothesis, I suppose. The way this concept is written, makes it seem such inconclusiveness has been already proved.

We agree and in the revision of the text, we have reformulated this section of the introduction and removed that specific sentence.

4. Lines 98-100. Aim n. 3 is not clear. Moreover, it seems it has not been addressed in the paper.

We agree that aim 3 was not clearly defined. We have tried to rephrase this now, by replacing aims 3 and 4 with the following aim:

“iii) to discuss the implications of these developmental changes in W_i on estimates of responses to atmospheric CO_2 . “

5. Lines 127-130. Not clear. This concept needs to be explained more in detail.

We changed the wording here to explain the concept better, now saying “As mentioned these approaches reconstruct historical changes in W_i by looking backwards in time using individual tree ring series of big trees. Thus, they implicitly include the full developmental trajectory for each tree growing from seedling into a large adult tree.”

6. Lines 142-144. This statement is not fully supported by Fig. 2 and Table 2. Most analyzed predictive variables have $p < 0.001$ level of significance. See for example the value of R-squared for the predictive variables “diameter” and “crown illumination”: is the difference between 0.499 and 0.515 enough to state that crown illumination has a stronger effect than diameter? Probably, from a statistical viewpoint, it can be correct, but what about the biological meaning? Moreover, a value of R-squared equal to 0.131 is considered very highly significant (even more, a value of 0.053 is significant too): this is probably due to a very high number of observations, but I believe values so close to zero mean no relation at all but only a statistical trick (this also applies to table 1). Moreover, the same information should not be reported both in figure and table.

We agree that a statistical comparison of r-squares 0.499 and 0.515 is very inconclusive, and indeed meaningless.

What we are arguing here is that the results indicate that ageing is not the real cause for the developmental change in W_i . This is because –and as we say in the results- W_i is generally more strongly related to crown illumination and height rather than age. In particular, for *Quercus* and *Fagus*, age has lower r-squared values (0.35 and 0.33) compared to crown illumination (0.51 and 0.54). In addition, the most parsimonious multiple regression model for each species consistently excludes age, as it does not improve the model. The outcome of the linear mixed effects model with all four species included also shows that height and crown illumination are the best predictors (in almost equal proportion).

Thus, we do find confirmation that age really is not the real driver for developmental trends - as expected really. We do not try to argue though that we can separate the other effects fully, and do acknowledge this in the discussion by saying “As tree height and crown illumination increase simultaneously in trees over the course of their life, it is impossible to fully separate the effect of the two variables with our methods.”

To avoid double reporting of R-squares, we now deleted the r-squares from figure 2.

7. In general, it is not clear which information is derived from data gained from this experiment and which from meta-data.

Please see our response to point 10.

8. The discussion is too long, sometimes speculative (e.g. lines 270-281) and some parts seem a sort of rephrasing of the introduction. It should be reduced in length and better focused on results. Apart from stating which are the bias in published papers, some solutions should be also suggested.

We agree that the discussion could have been better focused and that there was overlap with the introduction. In this revision, we improved the discussion as follows:

- We removed the speculative section 270-281.
- We shortened the overall length.
- We significantly revised the wording and overall flow of the discussion (as in other parts of the ms).
- Solutions to the bias are not straightforward, but we have added a reference to one possible bias correction for RCS approaches.

9. Lines 144-150. This is not straightforward. Please, check.

We rephrased this section of the results, and hope the text is now clear. Please see new text.

10. Methods are somewhat confusing. Are all data presented derived from previously published papers? Please refer also to comment n. 7. A summary information about the study sites, climate, plant age and features of the stands should be added. Were the tree rings dated only visually, without any cross-dating? To clarify this, also in response to point 7, we now added to the SI table 1 information about our sampling sites, including specific numbers of trees sampled and climate and vegetation of the sites.

To also further make clear that most of our data are newly collected, we made the following changes in the text:

In the introduction (lines 88-91): “Therefore, here we **collect new data** using a systematic size-stratified approach that controls for CO₂ to quantify, ...etc. ”

In the methods, we rephrased the first methods sentence to “In this study, we focus on four different species, *Pinus sylvestris* (Scots pine), *Quercus robur* (pedunculate oak), *Fagus sylvatica* (common beech), and *Cedrela odorata* (Spanish cedar) for which we present new and original data and compiled literature data (see SI table 1).”

We also added the following details on crossdating/age determination of the trees, including a reference for the annual character of *Cedrela* : “All four species form clear and annual rings, including the tropical *Cedrela* [Baker et al. 2016], and tree ages were estimated by ring counting without crossdating.”

[reference: Baker, J.C.A. et al. "Oxygen isotopes in tree rings show good coherence between species and sites in Bolivia." *Global and Planetary Change* 133 (2015): 298-308.]

11. Line 387. “When possible” is not acceptable. Please, quantify.

We have now specified this and changed the phrasing into “Increment cores were taken from at least two radii for *Quercus*, *Fagus* and *Pinus* and generally in three direction for *Cedrela*. Tree ring analysis on discs for *Cedrela* always based on at least 3 radii.”

12. The authors report that isotope analyses were done on the last 5 rings/years. Since the plants were not all collected in the same year (lines 365-394), does this mean that tree rings from different calendar years were analysed and compared?

Yes, this means that the rings from the two different sites and different calendar years were mixed. This was only the case for *Cedrela* however, and the dates for the two sites were only 9 years apart (i.e., in 2002 and 2011). We find that Wi-trends with age do not differ between the sites, as can be seen from the figure below (2009 site = black, and 2011 = red). Thus, it is justified to use data from both sites in one single analysis. We added to the first section on “isotope analysis” a sentence on this (see below).

The paragraph from line 408 is confusing. Please explain better the sampling.

We now rephrased the first section of this paragraph to better explain this. Specifically, we changed the wordings as follows: “We isolated and analyzed bulk samples containing the last five rings for *Fagus*, *Quercus* and *Pinus*. For *Cedrela*, we analyzed the isotope ratio for each ring over the last five years, and then took the average of those years. For *Cedrela* the sampling date differed by 9 years between the two sites that were used, but this did not affect the results, as we find no difference in W_i between these periods (data not shown).”

13. It is not clear to what extent the other environmental drivers and intrinsic plant traits (diameter, height and crown illumination) affect estimation in water use efficiency.

We are not sure what the reviewer exactly means here. We unraveled the various drivers as far as possible given the environmental and trait data that we decided to collect in the field. Based on these data we are able to show that for 3 species a large proportion of the variation is explained (i.e., between 60 and 78% for the broadleaf species). We feel this is quite nice, but of course there is always more data which could be collected.

Reviewer #2 (Remarks to the Author):

Tree-ring stable carbon isotopes have been used to infer CO_2 effects on intrinsic water-use efficiency of growing trees. This manuscript claims that such studies should account for confounding due to tree age and height, which increase in parallel with the recent rise in atmospheric CO_2 . The argument has been made before, but it is still frequently ignored, even in some high-profile, recent papers. This issue was brought into high relief by Keenan et al.'s nature paper in 2013, which used eddy flux to infer extreme increases in water-use efficiency in recent decades, dismissing the tree-ring isotope data, which shows smaller effects, as ridden with artifacts. This paper describes one of those artifacts and provides a means of correcting for it. Perhaps this will be seen in the future as a step toward reconciliation of tree-ring and

eddy-covariance techniques.

Specific comments: Change title: "Developmental trends in tree-ring carbon isotopes affect estimates of water-use efficiency responses to CO₂"

Thanks for the suggestion. Since developmental trends also impact effects on W_i by climate we change the title to: "Tree life history strongly affects estimates of water-use efficiency responses to climate and CO₂ from isotopes"

Lines 60 to 69: I appreciate this summary of "developmental effects," but I would present them differently. I suggest presenting the height effect first because it should be universal, controlled by the gravitational effect on water potential. In contrast, many trees do not grow under the canopies of other trees, but rather grow in even-aged stands. The beech studied here does grow under canopies, but the pine studied here does not. The light effects might thus be expected in the beech, but not the pine. Similar effects might be expected in the refixation of respired CO₂, which is probably more significant for trees growing up under the canopy than for short trees growing up after a disturbance. Refixation had minimal effects in an even-aged conifer stand with an open canopy (Brooks et al., 1995), which is again more similar to what one would expect in the Finnish pine trees studied here.

We followed the suggestion. The sentence now reads:

"as trees grow older they increase in height, which imposes gravitational constraints on water transport to leaves in the upper canopy, affecting stomatal conductance²⁶⁻²⁸. In addition, trees in closed-canopy forests experience strong increases in irradiance levels from the understory to the canopy, which affect rates of photosynthesis²⁹. These changes in stomatal conductance and photosynthesis as trees grow, both affect plant isotope discrimination^{30,31},"

Note also that after discussing these three effects, much of the results section deals with "age" as a predictor variable. I understand that age can be measured more precisely, but I advise that the section describing alternative causes should be concluded by a statement that age is what you could actually measure.

We already mention the age effect various times before (i.e., line 57 "giving rise to an age effect", line 60 "as trees grow older") and thus feel it would be too much to mention it here again.

Line 70: "very few" studies have looked at this before? It seems to me that this gives insufficient credit to previous work, some of which laid out these issues in detail.

That is fine. We now change the text and added credit to one of the earlier studies that explicitly assessed the magnitude before as follows.

"..., not many studies have explicitly assessed the magnitude of these effects (but see Marshall and Monserud 1996)."

Line 118, Fig. 1, and elsewhere: the authors express $iWUE$ in per mil when it should be $\mu\text{mol/mol}$ or ppm. This is confusing because the units of isotopic composition are per mil. This must be corrected.

Thanks for pointing this out and yes of course the units should be parts-per-million, ppm (or $\mu\text{mol/mol}$). We have corrected this throughout.

Line 226-229: not convinced you can use the light environment of similar-sized trees to infer the past light environment of mature trees. Oak and especially beech trees can grow in either bright light or heavy shade. At least this approach should be better justified and perhaps it should be removed from the manuscript.

We may not have made here very clear what we mean. We compared contemporary small trees of similar size but exposed to different light conditions. We rephrased this part of the text:

"Nevertheless, for *Fagus* and *Quercus*, we collected and analysed samples from trees that were of similar

size (all small saplings), but which differed strongly in their light environment. W_i 's differed markedly revealing light exposure as an important driver of variation in W_i , irrespective of the trees' size. "

line 237: this was demonstrated well before reference 25. see Ehleringer et al. in 1986.
Thanks for the suggestion. We clearly missed this reference and have now added it.

line 245: as they say this effect is rather small. Why emphasize it in the introduction?
We mentioned this (rather briefly) in the introduction to provide a complete overview of proposed mechanisms, and because the contribution of soil respiration was one of the first mechanisms described to explain changes in $\delta^{13}C$ with tree height (see Medina & Mincin 1980, Van der Merwe & Medina 1991). We demonstrate that the effect is quite small, but still feel it needs to be introduced in the intro.

Line 253: An alternative is to present these species as examples of shade tolerant (beech), intermediately tolerant (oak), and intolerant (pine) species. This would add to the generality and show a better connection to the literature.

We agree that the differences could possibly be due to shade tolerance. The pattern is not very clear however, as the shade-intolerant *Cedrela* also shows very strong developmental trends, and differences could also rather be due to differences between Gymnosperms and Angiosperms, or other factors. We added the following sentence to the discussion in the section on homeostatic gas exchange strategies: "These differences may be related to taxonomy (conifers vs. broadleaf species), difference in shade-tolerance between species, or growing conditions (open sites vs. forests).."

line 265: missing reference
Thanks for pointing out. We have now corrected this.

line 340: note that this comment agrees with Voelcker et al.
Yes we agree and add this observation to the discussion
" , reminiscent of observations of gas-exchange responses to CO_2^{12} ."

line 345: but not reference 42, which shows constant W_i
We have added this reference here "(but see ref. 42),"

Reviewer #3 (Remarks to the Author):

I found the manuscript interesting in sorting out the effects of age, height and exposure, although I had thought some of these issues had been dealt with earlier. Given that this work is to be applied to analysis of the effects of eCO_2 and climate change, I found it surprising that there is no ref to the work of Schubert, who applied a unifying relationship between carbon isotope fractionation and pCO_2 within C3 land plant tissues towards interpreting terrestrial organic sediments in the fossil record (Schubert and Jahren, 2013, 2015; Cui and Schubert, 2016). This work led to ancillary studies designed to evaluate the confounding effects of pCO_2 and climate on tree-ring $\delta^{13}C$ values (Schubert and Timmermann, 2015; Trahan and Schubert, 2016).

We thank the reviewer for the interesting references, and now cite the Schubert and Jahren 2015 paper in the discussion.

" implications for use of tree ring $\delta^{13}C$ for ... palaeoclimate reconstructions [McCarroll and Loader 2004, Schubert and Jahren 2015]."

Intrinsic water use efficiency is introduced without real explanation. One has to wait for the Methods. But once there I could not understand the result. But just getting to this point is frustrating – wondering what is meant by line 91 ("lifetime patterns of W_i ").

We introduce the concept in the introduction (see lines 44-49; "An increasingly popular method of

..etc.”), along with a reference to the Farquhar et al. 1989 publication and our methods. Introducing here the full equations would break the flow of the ms too much. We understand that the mistake of the units caused confusion and that this may be the reason the reviewer did not understand the result?

With “lifetime pattern of W_i ” we meant the trends in W_i over a trees’ full life-span, but realize this is indeed unclear. We now changed this into:

“For *Pinus*, we furthermore study the change of W_i over the lifetime of individual trees from sub-fossil trunks from Finland. These have been recovered from lakes and grew their entire life under more or less constant pre-industrial CO_2 levels (see ³⁷).

Line 118 How does W_i have the units of per mil? If this the trend in $\delta^{13}C$ it is huge. Or does it just mean 3.5 to 7 %?

This was a mistake and we have corrected this now throughout the manuscript.

Line 237 Height effects: see

Givnish TJ, Wong SC, Stuart-Williams H, Holloway-Phillips M, Farquhar GD. 2014. Determinants of maximum tree height in Eucalyptus species along a rainfall gradient in Victoria, Australia. *Ecology* 95(11), 2991–3007

Thanks for the suggestions. We are aware of this publication, but the study mainly reports on variation in wood $\delta^{13}C$ for the tallest trees along a rainfall gradient. It is not directly relevant here.

Line 265 “Review of published isotope data by ref. shows indeed weaker trends with tree height for wood compared to leaves.” Is there a missing reference here?

Yes, we corrected this and added the reference (McDowell et al. 2011)

Line 441 Eq 2. There is an unnecessary (and unmatched) bracket here. The reference here should be Farquhar and Richards 1984.

We deleted the unnecessary bracket. The calculation of plant discrimination is also given in Farquhar et al. 1982, and we prefer to use that reference, to avoid too many refs.

Line 452 Eq3. This is Farquhar et al. 1982.

We now did add this reference.

I think I also derived Eq5. But that is not my main worry.

Line 480 “Note that we use the shorter permille notation (‰) as units for W_i which is the equivalent of the molar fraction ($\mu\text{mol mol}^{-1}$).”

Firstly the CO_2 concentrations are in mole fractions, not molar fractions. Secondly I was unsure at first what you meant by “the equivalent”. But re-reading it and the results given elsewhere in the manuscript it seems to me that you have equated the CO_2 gradient in $\mu\text{mol mol}^{-1}$ (parts per million) to something in per mil, which is parts per thousand. In other words unless there is a definition of W_i that is new (and therefore should have been introduced much earlier in the text), the results are out by a factor of 1000.

It was a mistake and should, of course, have been parts-per-million (ppm). Sorry for this and for the confusion it seems to have caused. We have corrected this throughout in the manuscript.

Line 57 You have equated W_i with $Ca-C_i$. Shouldn’t it be $(Ca-C_i)/1.6$?

We have changed this in the text.

Again the units cannot be parts per thousand.

We have corrected this.

Reviewers' Comments:

Reviewer #1:

Remarks to the Author:

The authors carefully followed the suggestions and the manuscript has been much improved in all its sections. The clarifications about methods and experimental approach also made the results more easily understandable.

Reviewer #2:

Remarks to the Author:

The authors have considered the comments received on an earlier draft and I agree with them that the manuscript is stronger. I ask that they consider three additional comments.

First, the title. "Life History" is often used in ecology to describe the timing of events controlling population behavior, e.g., age at first reproduction or maximum fecundity. That is clearly not what this paper is about. In the interest of putting this paper in the hands of the right readers, I suggest that they use "Tree Height" in its place. This would capture all the sources of variation associated with tree height and would present a clearer description of what was done.

Second, the abstract becomes muddy in the last three sentences. I would say: "In fact, developmental trends in broadleaf species were at least as large as the trends previously assigned to CO₂ and climate. Credible future tree-ring isotope studies require explicit accounting for species-specific developmental effects before CO₂ and climate effects are inferred." Or something like that.

Third, the discussion becomes a little chatty around line 300. I suggest that that paragraph, in particular, be tightened up.

Reviewer #3:

Remarks to the Author:

The manuscript is improved, for example with the per mil mistake corrected. However there are still errors and things that are unclear.

The abstract needs to be clearer about the final conclusion. Just what was the dependence of W_i on $\delta^{13}C_{air}$, once the developmental effects were corrected for?

Line 49 "...plant isotope discrimination ($D_{13Cplant}$) can be calculated (see methods), which, when assuming that water vapour pressure differences do not change, provides an estimate of changes in the ratio between assimilation and stomatal conductance (A/g_s). This ratio is called intrinsic plant water-use efficiency (W_i)." This is plainly wrong. There was no such assumption made about unchanging water vapour pressure differences when the concept was introduced, and there is none needed here.

Line 62 I think you mean 'confound' rather than 'compound'.

Line 179 "It shows that below-canopy variation in CO₂ and $\delta^{13}C_{air}$ contributes resp. 11, 14 and 20% to the increases in W_i with tree age". This is incorrect. It should not affect W_i provided the correct value is used for $\delta^{13}C_{air}$.

Line 292-4 I did not understand this sentence,

Line 339 Ref 24 was the first to discuss these possibilities.

Line 364 " As already mentioned various studies that do control for developmental effects in tree ring data report relatively large increases in W_i 8,40,62 (but see ref. 38), consistent with

ecosystem flux measurements⁷, and with $\delta^{13}\text{C}$ observations in paleo records and CO_2 enrichment studies¹². " Again, it would be nice if the corrected values of W_i in the present study were made more obvious.

Line 462 "Analysis of the Cedrela samples from Purissima at GFZ were converted to CO_2 ..." Cannot convert an analysis into CO_2 .

Reference for Eq 2 is Farquhar & Richards AJPP.

Line 510 "and assuming that the carbon flux into the leaf equals the assimilation rate when photosynthesis is at steady state." - I don't understand what this means.

Line 521 mole fraction, not molar fraction, since 'molar' means mol/L

Table 1- I did not understand the headings or numbers here.

Response to Reviewers' comments:

Reviewer #1 (Remarks to the Author):

The authors carefully followed the suggestions and the manuscript has been much improved in all its sections. The clarifications about methods and experimental approach also made the results more easily understandable. We thank the reviewer for his/her previous suggestions, and are glad that the approaches are now clearer.

Reviewer #2 (Remarks to the Author):

The authors have considered the comments received on an earlier draft and I agree with them that the manuscript is stronger. I ask that they consider three additional comments.

First, the title. "Life History" is often used in ecology to describe the timing of events controlling population behavior, e.g., age at first reproduction or maximum fecundity. That is clearly not what this paper is about. In the interest of putting this paper in the hands of the right readers, I suggest that they use "Tree Height" in its place. This would capture all the sources of variation associated with tree height and would present a clearer description of what was done.

We thank the reviewer for this suggestion, and agree it may cause confusion. We have taken over the suggestion, and changed the title.

Second, the abstract becomes muddy in the last three sentences. I would say: "In fact, developmental trends in broadleaf species were at least as large as the trends previously assigned to CO₂ and climate. Credible future tree-ring isotope studies require explicit accounting for species-specific developmental effects before CO₂ and climate effects are inferred." Or something like that.

Thanks for this good suggestion. We have changed this formulation.

Third, the discussion becomes a little chatty around line 300. I suggest that that paragraph, in particular, be tightened up.

The lengthy description in this section was due to our attempt to explain tree ring standardization, and its associated biases for non-tree ring audience. However, we agree it was too lengthy and have now cut some part, and refer interested readers to the relevant publications. This shortened the section and makes the paragraph more straightforward.

Reviewer #3 (Remarks to the Author):

The manuscript is improved, for example with the per mil mistake corrected. However there are still errors and things that are unclear.

The abstract needs to be clearer about the final conclusion. Just what was the dependence of W_i on C_a , once the developmental effects were corrected for?
We have made the abstract and conclusions clearer as suggested by reviewer 2.

The first purpose of our paper is to demonstrate that various factors strongly affect inferences of historical water use efficiency when reconstructed from tree rings. We highlight one specific effect which is the role played by tree height. Proceeding as suggested here and in the comment below, and correcting for the height effect alone, we would have to conclude that in two species (oak and beech), water use efficiency decreased over time. However, we know this is an artefact due to differences in growth condition of large trees when they were small compared to small trees now. We would find it therefore scientifically unsound to claim that we can estimate water use efficiency trends based on these data available to us. That is in fact a key message of the paper, which we elaborately discuss in the manuscript.

Line 49 "...plant isotope discrimination ($D^{13}C_{plant}$) can be calculated (see methods), which, when assuming that water vapour pressure differences do not change, provides an estimate of changes in the ratio between assimilation and stomatal conductance (A/g_s). This ratio is called intrinsic plant water-use efficiency (W_i)."

This is plainly wrong. There was no such assumption made about unchanging water vapour pressure differences when the concept was introduced, and there is none needed here.

Yes, this was a wrong statement, and caused by a mix-up of the various versions of the manuscript. We have taken out the water vapor pressure statement and we thank the reviewer for pointing this out.

Line 62 I think you mean 'confound' rather than 'compound'.
We corrected this.

Line 179 "It shows that below-canopy variation in CO_2 and $\delta^{13}C_{air}$ contributes resp. 11, 14 and 20% to the increases in W_i with tree age". This is incorrect. It should not affect W_i provided the correct value is used for $\delta^{13}C_{air}$.

This is indeed incorrect. However, any calculation of W_i from tree ring isotope ratios constitutes an estimate as we did not measure $\delta^{13}C_{air}$ under the canopy for each tree. To make this clear, we added inferred to the sentence, which now reads:

“It shows that **inferred** increases in W_i with tree age for *Fagus*, *Quercus* and *Cedrela* are resp. 11, 14 and 20% lower when using below-canopy values of CO_2 and $\delta^{13}\text{C}_{\text{air}}$ (see SI Fig. 4).”

Line 292-4 I did not understand this sentence,

We have now changed the formulation, as follows (additions in bold and underlined):

Surprisingly, for both *Fagus* and *Quercus*, we find that time trends in W_i for the dominant trees (**by looking backwards in time using individual tree ring series of big trees, Fig. 1 right panels**) are much weaker than the increase in W_i with tree age for the same species from the same sites (**Fig. 1 left panels**).

Line 339 Ref 24 was the first to discuss these possibilities.

We added that reference.

Line 364 “As already mentioned various studies that do control for developmental effects in tree ring data report relatively large increases in W_i 8,40,62 (but see ref. 38), consistent with ecosystem flux measurements⁷, and with $\delta^{13}\text{C}$ observations in paleo records and CO_2 enrichment studies¹². “ Again, it would be nice if the corrected values of W_i in the present study were made more obvious.

Please see our response above.

Line 462 “Analysis of the *Cedrela* samples from Purissima at GFZ were converted to CO_2 ...” Cannot convert an analysis into CO_2 .

We have changed this, saying now: “*Cedrela* cellulose samples ... were converted to “

Reference for Eq 2 is Farquhar & Richards AJPP.

We have added this reference.

Line 510 “and assuming that the carbon flux into the leaf equals the assimilation rate when photosynthesis is at steady state.” - I don’t understand what this means.

We took out the section .. “when photosynthesis is at steady state”.

Line 521 mole fraction, not molar fraction, since ‘molar’ means mol/L

We have changed this.

Table 1- I did not understand the headings or numbers here.

We have made the following changes to the table:

Table caption:

“Analysis of age- and height-related changes in W_i for different life stages”

Table headings we now say:

“ W_i vs. Age trends, ppm (100yr)⁻¹”

And we have added Age to the subheadings (i.e.. Age > 25 yrs)

Similar changes were made to the height part of the table. We hope this fixed the issue.

Reviewers' Comments:

Reviewer #2:

Remarks to the Author:

The authors have made good use of the review comments. This manuscript make an important point and makes it forcefully. This problem has been simmering in the background for a long while. This manuscript will move it into the foreground.

Reviewer #3:

Remarks to the Author:

Some of my comments have been addressed.

The manuscript is still a little glib going from $\delta^{13}C$ to Intrinsic Water Use Efficiency, rather than including the variable contribution of mesophyll conductance, but the whole field does it so that one cannot pick on (criticise) this manuscript alone.

Line 51 Make clear that it is the stomatal conductance to the diffusion of water vapour to which you refer (otherwise it could easily be assumed that you mean stomatal conductance to the diffusion of CO_2 , particularly since you refer to the result in ppm, which is the usual notation for CO_2).

Line 496. 'we choose not to correct for leaf-to-wood offsets.' But mesophyll conductance is hardly a leaf to wood issue. It is intrinsically part of the photosynthetic process. and so it needs to be clearly acknowledged that the term in $(C_i - C_c)/C_a$ is being ignored (like everybody else in tree-ring research does).

line 503 Again, as I pointed out last time 'assuming that the carbon flux into the leaf equals the assimilation rate' is a mysterious phrase. I am unsure what the authors mean. Are they allowing for some tiny flux of carbon going up the xylem or something? Or is it the very old discussion about uptake not necessarily equalling assimilation until some biochemical changes take place? Or?..If it is the latter it would need a reference. But I'd really recommend deleting it.

Line 870 I have previously corrected this mistake. It is O'Leary, not Oleary.

Minor points

Line 86 The few studies THAT

line 275 'Notwithstanding' needs something to go with it. eg 'Notwithstanding this conclusion,'

Response to review

Below we set out how we responded to specific comments from Reviewer 3

Some of my comments have been addressed.

The manuscript is still a little glib going from $\delta^{13}\text{C}$ to Intrinsic Water Use Efficiency, rather than including the variable contribution of mesophyll conductance, but the whole field does it so that one cannot pick on (criticise) this manuscript alone.

Line 51 Make clear that it is the stomatal conductance to the diffusion of water vapour to which you refer (otherwise it could easily be assumed that you mean stomatal conductance to the diffusion of CO_2 , particularly since you refer to the result in ppm, which is the usual notation for CO_2).

We have added this to the text (i.e., “stomatal conductance for water vapor”)

Line 496. 'we choose not to correct for leaf-to-wood offsets.' But mesophyll conductance is hardly a leaf to wood issue. It is intrinsically part of the photosynthetic process. and so it needs to be clearly acknowledged that the term in $(C_i - C_c)/C_a$ is being ignored (like everybody else in tree-ring research does).

That is correct and we added this to the wording. “We choose not to correct for the mesophyll conductance term or leaf to wood offsets “

line 503 Again, as I pointed out last time 'assuming that the carbon flux into the leaf equals the assimilation rate' is a mysterious phrase. I am unsure what the authors mean. Are they allowing for some tiny flux of carbon going up the xylem or something? Or is it the very old discussion about uptake not necessarily equalling assimilation until some biochemical changes take place? Or?..If it is the latter it would need a reference. But I'd really recommend deleting it.

We have deleted it.

Line 870 I have previously corrected this mistake. It is O'Leary, not Oleary.

We have corrected this.

Minor points

Line 86 The few studies THAT

We have corrected this.

line 275 'Notwithstanding' needs something to go with it. eg 'Notwithstanding this conclusion,'

We have changed into “Notwithstanding these results. ... “